# Tumour-Derived, Extracellular Microvesicles in the Treatment of Acute Renal Failure: An Experimental Study

**DOI:** 10.3390/medsci13020035

**Published:** 2025-04-01

**Authors:** Galina V. Seledtsova, Victor I. Seledtsov, Ayana B. Dorzhieva, Irina P. Ivanova, Tatiana S. Khabalova, Adas Darinskas, Alexei A. von Delwig

**Affiliations:** 1Institute for Fundamental and Clinical Immunology, 630099 Novosibirsk, Russia; galina-seledtsova@yandex.ru (G.V.S.); dorzhieva-ayana@yandex.ru (A.B.D.); irinaiki@rambler.ru (I.P.I.); gvs1502@mail.ru (T.S.K.); delvig59@mail.ru (A.A.v.D.); 2Petrovsky National Research Centre of Surgery, 119991 Moscow, Russia; 3National Cancer Institute, 08406 Vilnius, Lithuania; darinskas.adas@gmail.com

**Keywords:** acute kidney injury, tumour, mesenchymal stem cell, microvesicles, regeneration

## Abstract

**Background/Objectives**: This investigation compared the therapeutic efficacy of extracellular microvesicles (MVs) derived from murine L929 sarcoma cells and murine mesenchymal stem cells (MSCs). **Methods**: A mouse model of acute kidney injury (AKI) was used. **Results**: Both MVs from L929 cells (L929-MVs) and MSCs (MSC-MVs), unlike those obtained from murine peripheral blood mononuclear cells (PBMCs), enhanced survival rates in AKI mice and significantly improved kidney function. This was indicated by decreased levels of urine albumin and serum creatinine. Furthermore, treatment with L929-MVs and MSC-MVs elevated the proportions of CD4+CD25+FOXP3+ regulatory T cells while reducing the presence of pro-inflammatory CD4+CD44+ T cells in the spleens of AKI mice. **Conclusions**: the results highlight the potential of tumour-derived MVs to facilitate organ repair and exert cytoprotective immunomodulatory effects.

## 1. Introduction

Extracellular vesicles (EVs) are membrane-enclosed particles released by all cell types. The loading of EVs with molecular components is a highly regulated process, involving the precise sorting of biomolecules. The formation and release of EVs occur in response to cellular activation or apoptosis, and they carry a variety of biomolecules such as proteins (such as tetraspanins, heat shock proteins, growth factors, cytokines, cell adhesion- and antigen presentation-related molecules, signalling and cytoskeleton proteins, transcription and protein synthesis-related proteins, metabolic enzymes, etc.) lipids (such as ceramide, cholesterol, sphingomyelin, phosphatidylserine, and saturated fatty acids), DNA (such as double-stranded DNA, single-stranded DNA, mitochondrial DNA, and circular DNA forms involved in horizontal gene transfer, cellular stress responses, and signalling), and RNA (mRNA, miRNAs, long noncoding RNA (lncRNAs), piwi-interacting RNA, and circular RNA) [1,2]. Based on their biogenesis, EVs are primarily classified into two categories: exosomes and microvesicles (MVs). Exosomes are formed through inward budding of the endosomal membrane, resulting in multivesicular bodies that eventually fuse with the plasma membrane, releasing the vesicles into the extracellular space. In contrast, MVs are produced through the outward budding of the plasma membrane, forming neck-like structures that pinch off to release vesicles. Exosomes typically range from 50 to 150 nm, while MVs span a larger size range of 100 to 1000 nm. Differentiating between exosomes and MVs is challenging due to overlapping size distributions, shared surface markers, and the absence of definitive identifiers [1].

The primary role of EVs is to facilitate the transport of biomolecules, protecting them from extracellular enzymes and other degrading factors. In addition to molecular cargo, MVs are capable of carrying cellular organelles, such as mitochondria. The stability of MVs is enhanced by their negatively charged surface and the CD47 biomarker, which helps them evade phagocytosis by mononuclear cells [3]. MV migration within the body is guided by the receptors expressed on their surfaces [4]. Evidence indicates that EVs can cross biological barriers, including the blood–brain barrier [1]. Acidic environments may induce the fusion of MVs with target cell membranes [5]. Current research supports the idea that EVs, including MVs, are key players in long-range intercellular communication and the regulation of adaptive cellular responses.

Mesenchymal stem cells (MSCs) are widely studied in regenerative medicine, with numerous clinical trials investigating their potential applications for various diseases [6,7,8]. Increasing attention has been directed toward EVs derived from MSCs (MSC-EVs), which have shown remarkable potential in immunosuppression, tissue repair, and disease treatment in preclinical models. MSC-EVs exhibit properties similar to those of MSCs themselves, including anti-inflammatory, anti-apoptotic, pro-angiogenic, and immunomodulatory effects [9,10]. Studies have highlighted the therapeutic benefits of MSC-EVs in experimental models of kidney injury, diabetic nephropathy, and kidney fibrosis [10]. Administration of MSC-EVs has been associated with significant reductions in biomarkers of renal damage, such as blood urea nitrogen, creatinine, and transaminase levels, along with improvements in necrotic lesions, tubular dilation, and cast formation [4,10,11].

This study provides the first evidence that MVs derived from tumour cells may aid in restoring kidney function in a murine model of acute kidney injury (AKI), potentially exhibiting effects comparable to those of MSC-MVs.

## 2. Materials and Methods

The research received approval from the Institutional Ethics Committee at the Institute of Fundamental and Clinical Immunology (Protocol No. 143, dated 29 November 2023).

### 2.1. Mice

Male CBA/J mice aged 4 to 6 months and weighing 18–20 g were used for all experiments. The animals were sourced from the breeding facility of the Goldberg Research Institute of Pharmacology and Regenerative Medicine (Tomsk, Russia). They were housed with access to autoclaved food and boiled water. All procedures involving animals complied with the legislation of the Russian Federation and Directive 2010/63/EU of the European Parliament and Council of 22 September 2010 on the protection of animals used in scientific research. Euthanasia was performed using cervical dislocation.

### 2.2. Generation of Bone Marrow-Derived MSCs

Bone marrow was harvested from the femurs and tibias of intact mice using a glass homogeniser. The cells were suspended in chilled RPMI 1640 medium. After washing, the cells were plated in plastic flasks containing a complete culture medium comprising RPMI 1640, 10% foetal calf serum, 2 mM L-glutamine, and antibiotics. All reagents were obtained from Sigma-Aldrich (St. Louis, MO, USA). Starting on day 3, non-adherent cells were gradually removed. The adherent mesenchymal stem cells (MSCs) exhibited a typical fibroblast-like morphology and formed a confluent monolayer within four weeks. The MSCs were detached using a 0.25% versene-trypsin solution and washed twice in serum-free medium.

### 2.3. L929 Cell Line

The L929 murine fibroblast cell line line was obtained from the N.N. Blochin Cancer Center (Moscow, Russia). L929 cells were cultured in RPMI 1640 medium supplemented with 10% foetal calf serum, 2 mM L-glutamine, and antibiotics. Like MSCs, L929 cells adhere to solid plastic surfaces and have similar morphology and size to MSCs.

### 2.4. Separation of Peripheral Blood Mononuclear Cells (PBMCs)

Peripheral blood mononuclear cells (PBMCs) were isolated by density gradient centrifugation using Ficoll-diatrizoate solution (density: 1.077 g/mL) at 1500 rpm for 40 min. The collected cells were washed, counted, and prepared for use in subsequent experiments.

### 2.5. Obtaining and Characterization of Apoptotic Extracellular MVs

Apoptosis in MSCs, L929 cells, and PBMCs (1 × 10⁶ cells/mL) was induced by incubating the cells in serum-free medium under oxygen-deprived conditions for 48–72 h. Hypoxia and nutrient deficiency are potent stimuli for cellular EV generation and secretion [12]. Following incubation, cell debris was removed by centrifugation at 2000× *g* for 15 min, and the supernatants were subjected to a second centrifugation at 12,000× *g* for 60 min at 4 °C to produce the MV pellet. For flow cytometric analysis, the MV pellet was resuspended in 100 μL of saline. Annexin V-positive MVs were analysed using a CytoFlex flow cytometer (Beckman Coulter Life Sciences, Indianapolis, IN, USA), with size calibration performed using TruCOUNT nanoparticles (TC) according to the manufacturer’s protocol.

### 2.6. Induction of AKI

AKI was induced by a single intramuscular injection of 50% glycerol at a dose of 200 µL/mouse [13,14]. This treatment led to rhabdomyolysis, which had a combined ischemic, toxic, and retention effect on the kidneys. Preliminary experiments showed that this procedure caused acute necrosis of the proximal renal tubule epithelial cells and a marked increase in blood serum urea levels.

### 2.7. Treatment of AKI Mice with MVs

MV preparations (derived from 2 × 10⁶ cells or approximately 50–100 µg per mouse) were administered intravenously into the retroorbital sinus one day following AKI induction. Mice with induced AKI were euthanised on the 11th day after MV administration. The study involved the following groups, each consisting of ten male mice:Intact miceControl mice with AKI, no treatmentAKI mice treated with MSC-derived MVs (MSC-MVs)AKI mice treated with MVs derived from L929 cells (L929-MVs)AKI mice treated with MVs derived from PBMCs (PBMC-MVs)

At least two independent, identical experiments were conducted.

### 2.8. Biochemical Analysis

Blood creatinine levels were quantified using the BioVision Creatinine Mouse ELISA Kit (rev 03/18, cat. No. E4369-100, Abcam, Cambridge, UK). The levels of fatty acid binding protein-1 (FABP1) in the blood were assessed using the Mouse/Rat FABP1/L-FABP Immunoassay Quantikine^®^ ELISA (cat. No. RFBP10, R&D Systems, Minneapolis, MN, USA). Urine albumin concentrations were determined using the Mouse Albumin ELISA Kit (cat. No. ab108792, Abcam, Waltham, MA, USA).

### 2.9. Histological Examinations

Murine kidneys were fixed in a 4% formalin solution, followed by standard steps of tissue processing for histopathology, including dehydration and paraffin embedding. Paraffin blocks were cut into 4–5 μm thick sections using a rotary microtome (Microm HM 340E; Carl Zeiss, Oberkochen, Germany) and stained with haematoxylin/eosin, Sirius red, or according to Mallory’s trichrome staining protocol. Light microscopy and microphotography were performed using a light microscope Axioskop 40 (Carl Zeiss, Oberkochen, Germany). Morphometric analysis of the histological kidney structure was conducted on paraffin sections by measuring the following morphometric parameters: diameters of superficial renal glomeruli, diameters of renal collecting tubules, and the size of cells in the middle third of the medullary region (ocular lens, 10 × 25; objective lens 63×).

### 2.10. Flow Cytometry

The spleens of mice were dissected into small fragments using scissors and then gently homogenised in cold versene solution using a glass homogeniser. The resulting cell suspension was allowed to settle to remove larger cell aggregates. After centrifugation, the cells were collected, washed, counted, and stained with anti-murine CD4-FITC, CD44-PE, CD4-APC, CD25-FITC, and FoxP3-PE antibodies (BD Biosciences, San Jose, CA, USA). Flow cytometric analysis was performed using a BD FACSCanto™ II Flow Cytometer (BD Biosciences, Heidelberg, Germany) or a CytoFLEX system (Beckman Coulter, Indianapolis, IN, USA).

### 2.11. Statistics

Data analysis was conducted using GraphPad Prism 8 (GraphPad Software, Boston, MA, USA) and one-way ANOVA. Tukey’s multiple comparisons test was used to validate the data. A *p*-value of <0.05 was considered statistically significant.

## 3. Results

### 3.1. Size Distribution of Isolated EVs

Flow cytometry analysis indicated that the isolated apoptotic EVs were predominantly larger than 100 nm in diameter, suggesting that the employed methodology primarily resulted in the isolation of extracellular MVs (Figure 1).

### 3.2. The Effects of MVs on Functionality and Morphological Structure of the Damaged Kidney

Given that AKI development is associated with a dramatic increase in albumin excretion, we also analysed urine albumin levels in control and experimental mice. On day 12, albumin levels in the urine of untreated AKI mice were found to be more than 30 times higher than normal. Statistically significant reductions in albumin levels were observed in the urine of AKI mice treated with MSC-MVs or L929-MVs (but not PBMC-MVs) (Figure 2).

Induction of AKI resulted in a 1.5-fold increase in blood creatinine levels (from 2.24 ± 0.17 to 3.11 ± 0.39 ng/mL, *p* < 0.05). Data presented in Figure 3 show that treatment of AKI mice with MSC-MVs or L929-MVs (but not PBMC-MVs) led to near-normalisation of serum creatinine levels.

Overall, treatment with L929-MVs or MSC-MVs led to a significant improvement in kidney function in AKI mice. Furthermore, our data indicate that both tumour-derived MVs and MSC-derived MVs demonstrated similar regenerative effects.

In the kidney glomeruli of untreated (control) AKI mice, the urinary space was almost undetectable, and there was pronounced glomerular dystrophy: The size of mesangial cells and the height of the collecting ducts were reduced (duct height 2.7 ± 0.2 μm). These data were significantly different from those of intact mice (duct height 4.04 ± 0.52 μm). Compared to control AKI mice, AKI mice treated with L929-MVs demonstrated an increase in the height of the collecting ducts to 3.6 ± 0.18 μm. The size of the renal glomeruli in control AKI mice and those treated with MSC-MVs or L929-MVs showed no significant differences. However, in the MV-treated groups, there was clear preservation of cellular structures. As shown in Figure 4, the administration of MSC-MVs and L929-MVs significantly reduced the severity of observable dystrophic changes in the kidneys of AKI mice.

### 3.3. Both L929-MVs and MSC-MVs Promote Cytoprotective Immunomodulation

The immune system constantly regulates regenerative processes in the body through both cytodestructive and cytoprotective immune responses, with T cells playing a critical role in determining the nature and magnitude of these reactions. To investigate the effects of regenerative MVs, we analysed the proportions of CD25+FOXP3+ and CD44+ T cells within the CD4+ T cell population in the spleens of AKI mice. Regulatory CD4+ CD25+FOXP3+ T cells are well-known for their potent anti-inflammatory functions, which help to down-regulate immune-mediated cytodestructive responses [15,16]. In contrast, CD44+ T cells are commonly linked to pro-inflammatory immune activity, including the pathological inflammation seen in chronic disease models [17]. Our findings indicate that treatment with L929-MVs or MSC-MVs in AKI mice resulted in a significant increase in cytoprotective CD4+ CD25+FOXP3+ T cells in the spleen (Figure 5) as well as a decrease in cytodestructive CD4+ CD44+ CD62L+ T cells (Figure 6).

Overall, our findings suggest that one potential mechanism driving the regenerative potential of MVs may involve a systemic decrease in pro-inflammatory, cytodestructive immune responses resulting from tissue damage.

## 4. Discussion

The tumour is recognized for its pathological regenerative activity, which enables it to modify its immediate microenvironment to support its own growth. Tumour-derived EVs play a pivotal role in this process by carrying biologically active molecules that stimulate tumour cell growth. These vesicles interact with immune cells, suppressing anti-tumour cytotoxic responses and reprogramming systemic immunity to aid tumour invasion [18]. Interestingly, both tumour and normal tissues utilise similar mechanisms and biomolecules to regulate regenerative processes [19]. In this study, we conducted a comparative analysis of tumour-derived MVs and MVs from bone marrow-derived MSCs to assess their effects on kidney function in a murine model of AKI. Previous research has shown that MSC-derived EVs, including MVs, possess regenerative activities comparable to MSCs themselves [9]. These EVs have demonstrated efficacy in reducing reactive oxygen species (ROS) production in renal tubular epithelial cells under hypoxic injury conditions and have been shown to reduce serum markers of kidney failure, such as urea and creatinine, while promoting renal tissue regeneration [11]. The therapeutic effects of MSC-derived EVs are thought to be partly mediated by small non-coding microRNAs and transforming growth factor β, which enhance the functional activity of regulatory T cells [9]. Exosomes and MVs differ not only in size but also in their biogenesis, membrane composition, and cargo profile. Due to their smaller size, exosomes are enriched in nucleic acids (e.g., miRNAs, lncRNAs) and specific protein markers associated with endosomal sorting, while the larger MVs can carry a broader range of cellular components, including larger RNA species, organelle fragments, and even whole ribosomes. This size difference influences their functional roles, with exosomes often implicated in fine-tuned intercellular communication and MVs in broader physiological and pathological processes [1,2,20].

While both MVs and exosomes have regenerative properties [1,2], we propose that MVs offer distinct advantages over exosomes as therapeutic agents. First, MVs, like cells, contain receptor molecules on their surfaces that enable them to migrate toward and interact with target cells. Second, MVs have the ability to transfer not only biomolecules but also cellular organelles, such as mitochondria, which are typically not viable in the extracellular space. Research has shown that MVs can deliver functional mitochondria into target cells [21], enhancing cellular energy production and likely contributing to the regenerative effects observed in damaged tissues [22].

Our findings suggest that tumour-derived MVs may have significant regenerative effects on the kidney, potentially rivalling or even surpassing those of MVs originating from bone marrow-derived MSCs. In particular, our experiments revealed that MVs derived from non-kidney-origin tumours, such as the L929 sarcoma cell line, were able to stimulate kidney tissue regeneration. This led us to hypothesize that tumour-derived MVs, regardless of their origin, may possess the capacity to regenerate not only kidney tissues but also other damaged organs, much like MSC-derived MVs. Our data from a chronic kidney injury (CKI) model support this hypothesis, demonstrating that MVs derived from various tumours, including L929 sarcoma, LLC carcinoma, and B16 melanoma, exhibited similar regenerative potential in the kidney [23]. It is well known that PBMCs do not possess the regenerative properties of MSCs, nor do they have the unlimited growth potential observed in tumour cells. Unlike MSCs, which have differentiation potential and contribute to tissue repair, PBMCs mainly function in immune responses. Therefore, in our experiments, PBMC-MVs were used as an inactive negative control. As expected, PBMC-MVs did not exhibit significant regenerative properties for the damaged kidney.

In our study, both L929-MVs and MSC-MVs significantly improved kidney function in the AKI model, where regenerative cell proliferation is limited. We hypothesize that in such settings, MV-mediated mitochondrial transfer could provide substantial energy support to the damaged organ, aiding in its functionality and subsequent regeneration.

Published data indicate that both MSCs and tumour cells, along with their EVs, exhibit immunosuppressive properties contributing to maintaining immune tolerance and preventing autoimmune reactions. These properties primarily inhibit cytodestructive immune responses, in part by enhancing the functional activity of regulatory CD4+CD25+FoxP3+ T cells. Regulatory T cells are essential for creating an immunological environment conducive to regenerative processes in both normal and pathological (tumour) tissues [15,16,24]. Our study found that both L929-MVs and MSC-MVs were effective in increasing the relative abundance of Tregs and decreasing the levels of pro-inflammatory CD4+CD44+ T cells in the spleens of AKI mice. While further investigation is needed, these data strongly suggest that immunomodulation plays a significant role in the regenerative effects observed with both tumour-derived MVs and MSC-MVs.

From a practical and cost-effectiveness standpoint, generating MSCs is labour-intensive and technically challenging, requiring well-characterized donor cells. In contrast, tumour cells can be easily cultured and expanded in vitro without the need for donor material. Furthermore, isolating MVs is simpler and less expensive than isolating exosomes, which typically require costly and complex procedures such as ultracentrifugation or microfiltration. MV preparations can be stored in frozen or lyophilized forms for long periods with minimal loss of activity [25], and such preparations can be standardised based on their physicochemical and biological properties [26].

Despite the promising regenerative potential of tumour-derived MVs demonstrated in this study, concerns about their oncogenic potential remain. Tumour-derived MVs have been shown to promote tumour growth [27]. However, it is important to note that these MVs do not contain directly mutagenic substances; they likely accelerate the growth of pre-existing malignant cells rather than inducing carcinogenesis in healthy cells. In our experimental model, a six-month follow-up of 14 mice treated with MV preparations revealed no increase in malignancy incidence, and none of the animals died from tumours. Nonetheless, the safety of long-term and repeated administration of tumour-derived MVs for treating chronic conditions, including age-related diseases, warrants further investigation. While concerns about the clinical application of tumour-derived MVs for life-threatening conditions persist, the therapeutic benefits may outweigh the oncogenic risks, as evidenced by the positive results in our AKI model. This consideration is especially relevant for patients with renal failure or other severe organ disorders, particularly those with high mortality rates.

Three criteria are needed to define MSCs: adherence to plastic, specific surface antigen (Ag) expression, and multipotent differentiation potential. In this particular study, these parameters were not determined, which can be considered a limitation of the study. However, it is worth noting that the most commonly used method for obtaining MSCs from bone marrow was employed in this research [28].

Finally, we stress that this paper represents a foundational phenomenological and descriptive step towards understanding the vast regenerative potential of tumour-derived MVs, which endows tumours with their hallmark immortality. We hope this study will serve as a catalyst for other studies to further explore this promising field. We have no doubt that essential future experiments designed to delineate potential underlying mechanisms and pathways, such as protein and gene expression profiling, will be conducted and reported in due course.

## 5. Conclusions

Our data suggest that tumour-derived MVs (but not PBMC-derived MVs), similar to MSC-MVs, may have the potential to enhance kidney function in a murine model of AKI. Notably, MVs derived from non-kidney-origin tumours stimulated the regeneration of damaged kidney tissues. This finding implies that tumour-derived MVs possess a multifactorial yet non-specific regenerative potential, likely linked to their immunomodulatory activity.

## Figures and Tables

**Figure 1 medsci-13-00035-f001:**
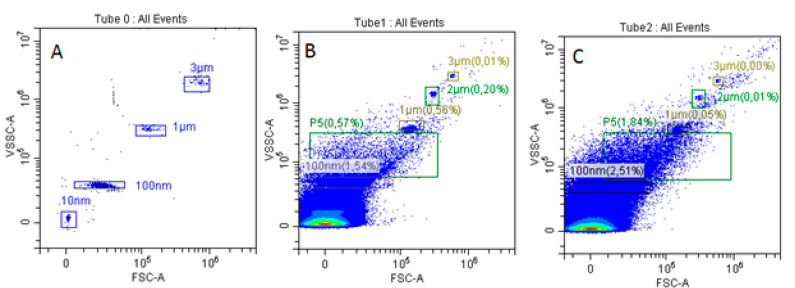
Analysis of exovesicle size distribution by flow cytometry. (**A**) Negative control without exovesicle-containing standard annexin V-FITC particles; (**B**) exovesicles derived from MSCs; (**C**) exovesicles derived from the L929 tumour cell line. Events above the threshold levels in the FITC channel were gated and analysed on SSC/FCS histograms; 10 nm—standard particles of 10 nm in diameter for gating microvesicles according to size; P5—area used for analysis and enumeration of microvesicles sized between 100 nm and 1 μm; P3—area used for analysis and enumeration of exovesicles sized 3 μm for normalizing exovesicle enumeration.

**Figure 2 medsci-13-00035-f002:**
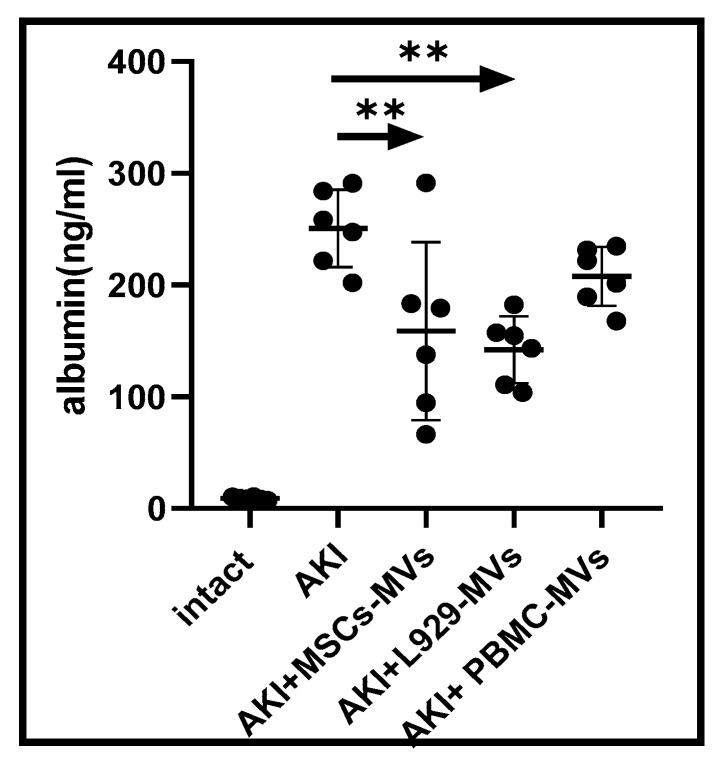
Albumin concentrations in the urine of acute kidney injury mice. Albumin levels were measured in untreated control acute kidney injury mice as well as in acute kidney injury mice treated with L929 microvesicles, mesenchymal stem cell microvesicles, or peripheral blood mononuclear cell-derived microvesicles. n = 6 ** *p* < 0.05 compared to control acute kidney injury mice.

**Figure 3 medsci-13-00035-f003:**
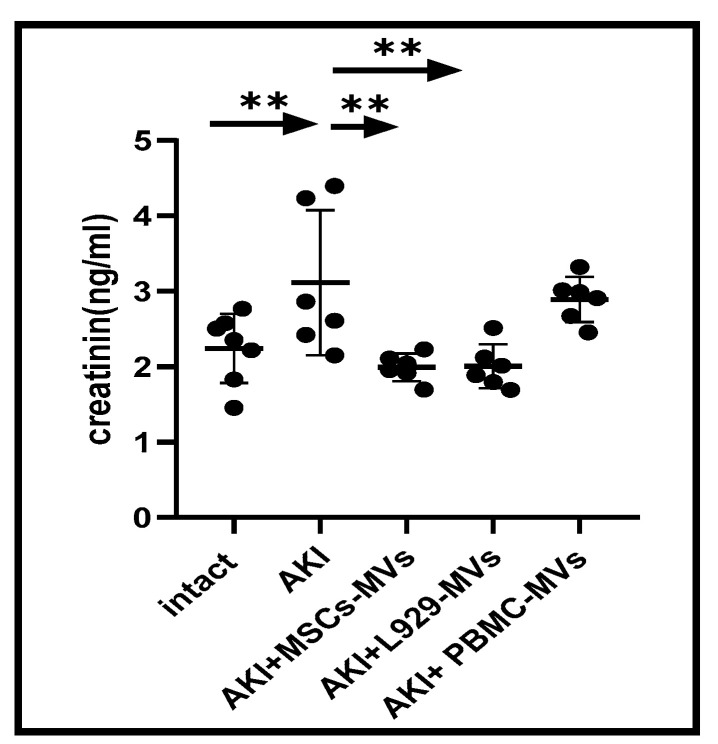
Creatinine concentrations in the sera of acute kidney injury mice. Serum creatinine levels were measured in untreated (control) acute kidney injury mice and in acute kidney injury mice treated with L929 cell-derived microvesicles, mesenchymal stem cell-derived microvesicles, or peripheral blood mononuclear cell-derived microvesicles. n = 6 ** *p* < 0.05 compared to intact or control acute kidney injury mice.

**Figure 4 medsci-13-00035-f004:**
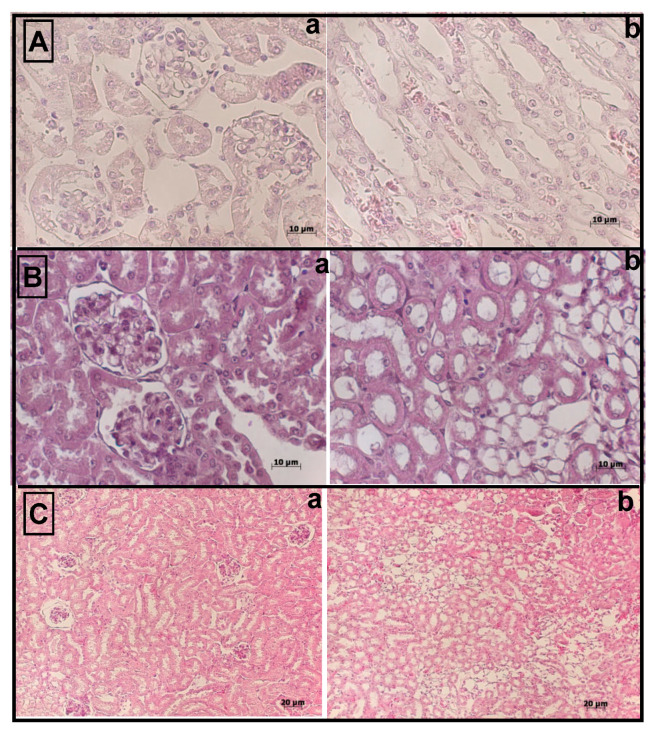
(**A**–**C**) Morphological structure of the damaged kidneys in acute kidney injury mice. (**A**) acute kidney injury mice, (**B**) acute kidney injury mice treated with mesenchymal stem cell-derived microvesicles, (**C**) acute kidney injury mice treated with L929 cell-derived microvesicles; (**a**) kidney glomeruli, proximal and distal tubules, (**b**) collecting ducts and Henle’s loops at the junction between the cortex and medulla of the kidney.

**Figure 5 medsci-13-00035-f005:**
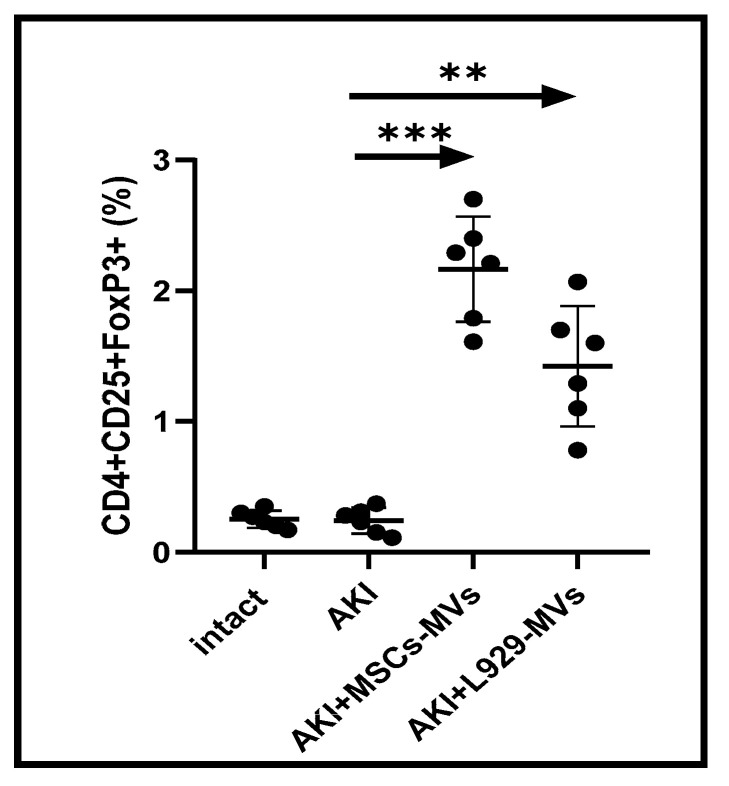
Relative proportion of cytoprotective regulatory T cells in the spleens of acute kidney injury mice. The percentage of CD4+CD25+FOXP3+ T cells was determined in the spleens of untreated control acute kidney injury mice and acute kidney injury mice treated with mesenchymal stem cell-derived microvesicles or L929 cell-derived microvesicles. Measurements were taken on day 12 following acute kidney injury induction. n = 6, *** *p* < 0.001, ** *p* < 0.05 compared to control acute kidney injury mice.

**Figure 6 medsci-13-00035-f006:**
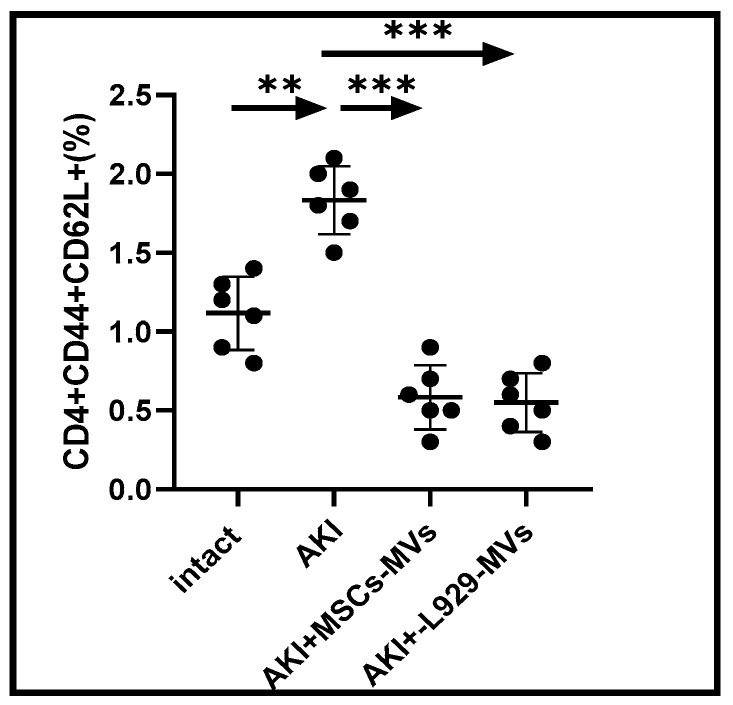
Relative proportion of cytodestructive pro-inflammatory T cells in the spleens of acute kidney injury mice. The percentage of CD4+CD44+CD62L+ T cells was assessed in the spleens of untreated control acute kidney injury mice and acute kidney injury mice treated with mesenchymal stem cell-derived microvesicles or L929 cell-derived MVs. Measurements were taken on day 12 after AKI induction. n = 6, *** *p* < 0.001, ** *p* < 0.05 compared to control or acute kidney injury.

## Data Availability

The authors do not have permission to share data.

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
