# Peer review of "Tumour-Derived, Extracellular Microvesicles in the Treatment of Acute Renal Failure: An Experimental Study"

_medsci, 2025, doi:10.3390/medsci13020035_

Round 1
Reviewer 1 Report
Comments and Suggestions for Authors
Authors of this comparative study tried to assess the therapeutic potential of extracellular microvesicles (MVs) derived from tumor (murine L929 sarcoma) cells and murine mesenchymal stem cells (MSCs) in an experimental animal model of acute kidney injury (AKI). The study revealed that both L929-derived MVs and MSC-derived MVs improved the survival of AKI animals and significantly restored kidney function confirmed by reduced albuminuria and serum creatinine levels. In addition, both L929-MVs and MSC-MVs increased the relative numbers of cytoprotective regulatory T cells and decreased the relative content of cytodestructive pro-inflammatory T cells in the spleens of AKI mice. However, these beneficial effects were not confirmed by MVs from murine peripheral blood mononuclear cells. They conlude, accordingly, that tumor-derived MVs may have the potential to promote the regeneration of damaged organs and induce cytoprotective immunomodulation.
The study describes a relatively new and inovative approach to acute kidney injury (AKI) management and deserves, therefore, attention. The article describes methods comprehensively. It is very appropriate that authors addressed concerns about potential oncogenic activity of extracellular microvesicles derived from tumor although during follow-up of mice treated with MV preparations an increased incidence of malignancy was not revealed.
However, there are some issues that should be addressed:
1. there are several types of AKI and its management depends on a cause. Some of them are easily defined although difficult to treat and it does not seem very likely that the same approach would suit to all of them. A more personalized approach would be more appropriate. In addition, there are several stages of AKI, according not only to albuminuria and serum creatinine levels but to urine output as well which was not addressed in this study. Authors should, therefore, discuss these limitations in more detail.
2.
3. Results
3.2. The effects of MVs on functionality and morphological structure of the damaged kidney
Given that AKI development is associated with a dramatic increase in albumin excretion, we also analyzed urine albumin levels in control and experimental mice. ......
- Comment: AKI development is not necessarily associated with a dramatic increase in albumin excretion. An example is acute tubulointerstitial nephrits where dramatic serum creatinine levels increases can be observed in the absence of associated increase in albumin excretion in urine.
3. Abbreviations in figures should be better explained in order to understand the meaning of figures, especially Fig. 1, which should be made more clear and concise. In addition, units are missing in some results, like this:
......................... The development of AKI led to a 1.5-fold increase in blood creatinine (from 2.24±0.17 to 3.11±0.39, units? P < 0.05). ................
4. Discussion
1st paragraph:
................ The tumor is recognized for its pathological regenerative activity, .............
- Comment: authors should better explain which tumors have these features - all of them or those evaluated in this study?
.................. Notably, MSCs and MSC-EVs have been demonstrated to reduce serum markers of kidney failure (such as urea, creatinine, and transaminases) and .....................
- Comment: transaminases are not serum markers of kidney failure but marker of hepatic failure. This should be corrected.
Author Response
Please see the attacment.

Reviewer 2 Report
Comments and Suggestions for Authors
This study aims to assess the therapeutic potential of extracellular microvesicles derived from murine L929 sarcoma cells and murine bone marrow-derived mesenchymal stem cells in a murine model of acute kidney injury. The topic is interesting and relevant. The manuscript is well-written, requiring a few revisions. Please address all the comments below:
Due to variations in the activity of MSCs from different sources, MSC nomenclature requires the source to be indicated at all times. Please revise to indicate that MSCs are bone marrow-derived throughout the paper.
Introduction/Discussion - Add a discussion on why using the microvesicles over the direct MSCs transplantation offers advantages by avoiding issues on storage, administration, and retention after delivery. Add references: https://doi.org/10.3390/ijms241411754
Introduction/Discussion - Add a discussion on the differences of exosomes and microvesicles apart from size. What is the effect of the size difference on the types of biological cargo? Add references: https://doi.org/10.1038/s41536-019-0083-6
Methods - Please include where the L929 murine fibroblast cells were obtained.
Methods - Three criteria are needed to define MSCs: adherence to plastic, specific surface antigen (Ag) expression, and multipotent differentiation potential. Did the authors perform any surface antigen characterization? Please add this as a study limitation. DOI: 10.1080/14653240600855905
Methods/Discussion - Tumors and MSCs naturally produced extracellular vesicles. Add an explanation on why the authors induced the production with hypoxia, and how might the contents of vesicles could differ with different induction methods.
Results - For all figures, define all abbreviations used in the caption.
Results - Figure 4. Annotate the pathologic findings (e.g. glomerular dystrophy) and/or useful histologic landmarks using arrows or arrowheads.
Discussion - The main concern with tumor-derived vesicles is the risk of malignancy. Add references for the statement " However, it is important to stress that these MVs do not contain substances that are directly mutagenic."
Author Response
Please see the attacment.

Reviewer 3 Report
Comments and Suggestions for Authors
The study presents a novel investigation into the therapeutic potential of tumor-derived extracellular microvesicles (L929-MVs) for acute kidney injury (AKI), offering a compelling comparison with mesenchymal stem cell-derived MVs (MSC-MVs). While the findings are intriguing, several issues should be addressed to strengthen the manuscript:
Major concerns:
The study lacks mechanistic analysis. Proteomic or RNA sequencing data identifying specific cargo components (e.g., miRNAs, proteins, mitochondria) in L929-MVs could improve the manuscript. Additionally, evidence of MV uptake by renal cells is also beneficial.
Minor concerns:
Define "Intact mice" explicitly in the Methods (presumably healthy, untreated control mice).
Clarify whether "Control mice with AKI without treatment" received vehicle injections or were untreated post-AKI induction.
The color differences between groups in Figure 4 raise questions about staining consistency. Specify the staining protocol (e.g., H&E, PAS) in the Methods, including fixation, sectioning, and imaging parameters.
Elaborate on MV isolation protocols to ensure reproducibility.
Address why PBMC-MVs failed to show efficacy. Are there differences in cargo or surface markers compared to L929/MSC-MVs?
Round 2
Reviewer 2 Report
Comments and Suggestions for Authors
All the issues have been addressed.
Author Response
Thank you very much for your positive review.
Reviewer 3 Report
Comments and Suggestions for Authors
At line 108, add "to produce the MV pellet" after "at 4°C" to avoid confusions.
As to why PBMC-MVs failed to show efficacy, it needs to be discussed in the manuscript. It is not clear to readers that why PBMC-MVs was investigated and how the results are related to the conclusion of this study.
Author Response
- At line 108, add "to produce the MV pellet" after "at 4°C" to avoid confusions.
- As to why PBMC-MVs failed to show efficacy, it needs to be discussed in the manuscript. It is not clear to readers that why PBMC-MVs was investigated and how the results are related to the conclusion of this study.
Our responses:
Thank you for your valuable comments.
- As suggested by the reviewer, we added the following statement to the revised MS: “to produce the MV pellet",.
- As suggested by the reviewer, we added the following statements to the revised MS (lines 294-300) “ It is well known that PBMCs do not possess the regenerative properties of MSCs, nor do they have the unlimited growth potential observed in tumor cells. Unlike MSCs, which have differentiation potential and contribute to tissue repair, PBMCs mainly function in immune responses. Therefore, in our experiments, PBMC-MVs were used as an inactive negative control. As expected, PBMC-MVs did not exhibit significant regenerative proper-ties for the damaged kidney”.